# Evaluate then Cooperate: Shapley-based View Cooperation Enhancement for Multi-view Clustering

**Fangdi Wang[1], Jiaqi Jin[1], Jingtao Hu[2],**
**Suyuan Liu[1], Xihong Yang[1], Siwei Wang[2],\*,**
**Xinwang Liu[1],\*, En Zhu[1],\***
[1] National University of Defence Technology
[2] Academy of Military Science
{wangfangdi19, wangsiwei13, xinwangliu, enzhu}@nudt.edu.cn

## Abstract

The fundamental goal of deep multi-view clustering is to achieve preferable task performance through inter-view cooperation. Although numerous DMVC approaches have been proposed, the collaboration role of individual views have not been well investigated in existing literature. Moreover, how to further enhance view cooperation for better fusion still needs to be explored. In this paper, we firstly consider DMVC as an unsupervised cooperative game where each view can be regarded as a participant. Then, we introduce the Shapley value and propose a novel MVC framework termed **S**hapley-based **C**ooperation **E**nhancing **M**ulti-**v**iew **C**lustering (**SCE-MVC**), which evaluates view cooperation with game theory. Specially, we employ the optimal transport distance between fused cluster distributions and single view component as the utility function for computing shapley values. Afterwards, we apply shapley values to assess the contribution of each view and utilize these contributions to promote view cooperation. Comprehensive experimental results well support the effectiveness of our framework adopting to existing DMVC frameworks, demonstrating the importance and necessity of enhancing the cooperation among views.

## 1 Introduction

Recently, multi-view clustering has become one of the most prominent problems in unsupervised learning[1, 2, 3, 4]. It leverages data from different views, combining them to provide richer information for clustering tasks. Generally speaking, multi-view clustering can be broadly categorized into traditional methods [5, 6, 7, 8] and deep multi-view clustering (DMVC) [5, 9, 10, 11, 12, 13, 14, 15, 16, 17, 18, 19] methods. Compared to the limited feature extraction capabilities of traditional methods, deep multi-view clustering methods better preserve information from different views through flexible deep neural networks. For most DMVC methods, the key challenge is how to enhance cooperation among views to accomplish better clustering performance[20].

Numerous DMVC algorithms have already achieved significant results in downstream clustering tasks. However, there has been little research on investigating the the contribution of each view in the fusion stage. Once evaluating the view contribution, we may observe the phenomenon: One view dominates the fusion process, suppressing the collaborative contribution of other views, leading to suboptimal clustering results. We attribute this phenomenon to insufficient cooperation between views, and thus propose a fundamental assumption: More balanced contributions and more extensive cooperation among multiple views can lead to better clustering results, as shown in Fig. 1.

---

*Corresponding author

38th Conference on Neural Information Processing Systems (NeurIPS 2024).

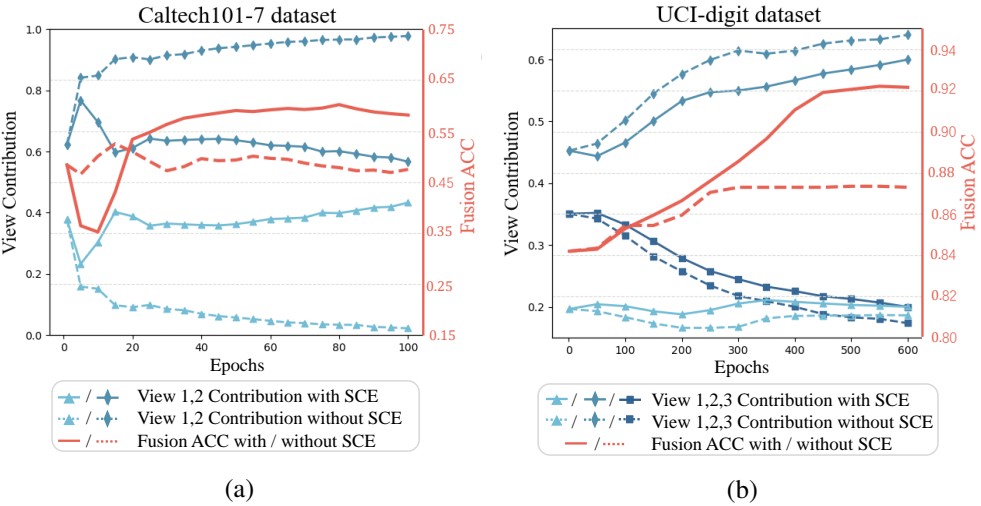

Figure 1: Performance evaluation of view cooperation contribution and fusion ACC on (a) Caltech101-7 (b) UCI-digit. As can be seen, our proposed SCE framework clearly promotes inter-view cooperation and achieves more satisfactory performance.

To this end, we consider introducing the concept of game theory into multi-view clustering and propose the **S**hapley-based **C**ooperation **E**nhancing (SCE) method. Shapley value[21, 22] is a concept used to measure the contribution of participants in game theory[23, 24]. It utilizes a utility function to evaluate the different marginal contributions that each participant brings when combined with other participants, and calculates their contribution to the overall cooperation by taking the weighted average over all possible combinations of participants[21]. Upon quantifying the contribution of each view, the cooperation among views can be analyzed based on their respective contributions. By employing a module that enhances view cooperation, we can dynamically adjust the parameter convergence rate of views during training in accordance with their contributions. This enables the underrepresented views to receive a more balanced training regimen, thereby enhancing post-fusion performance[25].

The primary contributions of this work can be summarized as follows:

• Our work is the first framework to evaluate the view contributions within an unsupervised context. Leveraging the proposed View Contribution Evaluation Module, we theoretically quantify the individual contributions of views.

• We propose the View Cooperation Enhancing Module based on the view contributions obtained by the View Contribution Evaluation Module, allowing suppressed views to effectively participate in the fusion process, ultimately enhancing clustering performance.

• Comprehensive experiments have been conducted to showcase the versatility of our SCE framework, which can effectively evaluate the contributions of views across various MVC frameworks.

## 2 Related Work

### 2.1 Deep Multi-view Clustering

In deep multi-view clustering[26, 27, 28, 29], deep neural networks[30, 31] with multiple nonlinear transformations are more effective at acquiring feature representations than traditional shallow models. Presently, deep multi-view clustering methods fall into three main categories[32, 33]: joint methods[34, 35, 36], alignment-based methods[14, 37, 38, 39], and other methods[40, 41, 42]. Joint methods consider differences and complementarities among views, optimizing sample representations within each view's space. For example, the DMJC[35] utilizes autoencoders to generate view representations, refining them as pseudo-labels for self-supervised optimization. Alignment-based methods focus on consistency between views, mapping representations into a shared subspace. These methods often employ contrastive learning, such as ProImp[43], which integrates contrastive learning

at both sample and prototype levels. Some approaches combine joint and alignment-based methods, like the share-specific method[40], which separates sample representations into shared and specific components, aligns shared representations into a common space, and refines specific representations in separate spaces.

## 2.2 Rethinking of Cooperation in DMVC

The core concept of DMVC is to enhance clustering performance by fostering enhanced cooperation and utilizing complementary information across views. Although existing DMVC research emphasizes leveraging inter-view cooperation for improved clustering tasks, there is currently a lack of research quantifying the contribution of views during the fusion stage to enhance cooperation. Multi views[20] and multi modalities[44] are two closely intertwined domains. Within the realm of multimodal fusion, studies have surfaced highlighting the issue of subpar collaboration among modalities, prompting endeavors to alleviate the imbalance in modality cooperation. Wang et al. [45] uncovered varying convergence rates among diverse modalities, leading to a scenario where jointly trained multimodal models struggle to match or outperform their unimodal counterparts. Furthermore, Peng et al. [25] demonstrated that in the context of audio-visual learning, superior performance in the audio modality hampers the optimization of the video modality.

Drawing inspiration from prior studies, our research aims to evaluate the individual contributions of each view in multi-view clustering fusion. Nevertheless, evaluating view contribution based solely on labels in unsupervised environments is infeasible. Moreover, employing weights as a measure for view contribution may not align with all DMVC structures, especially those based on contrastive learning frameworks. Consequently, quantifying view contributions in fusion and fostering their cooperation pose significant research challenges.

## 2.3 Shapley Value: a Method for Evaluating Contribution.

In order to reasonably and effectively evaluate the contributions of each view in the fusion process, we refer to the theoretical framework of game theory. The Shapley value [46] within game theory proves instrumental in quantifying participants' contributions to cooperative coalitions. Denote $\mathcal{X} = \{x_i\}_{i=1}^n$ as the alliances of $n$ participants. For a participant $x_i$, the Shapley value considers $x_i$'s contribution in every set that includes him. Let $S_i = \{S \subseteq \mathcal{X} | x_i \in S\}$ represent the set of all subsets of $\mathcal{X}$ that include the participant $x_i$, then, the overall contribution of member $x_i$, $i.e.$, the Shapley value, can be expressed as

$$Shapley_i = \sum_{s \in S_i} \frac{(|s|-1)!(n-|s|)!}{n!} [v(s) - v(s \backslash \{i\})], \tag{1}$$

where $|s|$ signifies the cardinality of set $s$; $v(\cdot)$ represents the utility function; $[v(s) - v(s \backslash \{i\})]$ quantifies the marginal contribution of $x_i$ to set $s$; and $\frac{(|s|-1)!(n-|s|)!}{n!}$ denotes the weight assigned to this marginal contribution, determined by the probability of set $s$ occurring.

Given the advantageous characteristics of Shapley values—efficiency, symmetry, dummy, and additivity—researchers have increasingly employed them for explaining machine learning models. For instance, Lundberg et al. [47] proposed a method to interpret predictions by computing feature importance based on Shapley values. Hu et al. [48] applied Shapley values to evaluate modality contributions in supervised tasks, while Wei et al. [49] explored modality assessment using Shapley values. However, these studies focus mainly on supervised scenarios, making the definition of utility functions for Shapley values in unsupervised settings a challenging task yet to be resolved.

# 3 Method

## 3.1 Problem Statement

Given a multi-view dataset $\mathcal{X} = \{\mathbf{X}^{(1)}, \mathbf{X}^{(2)}, ..., \mathbf{X}^{(V)}\}$ , where $V$ is the number of views, and $\mathbf{X}^{(v)}(v = 1, 2, ..., V)$ denotes the original feature space of the $v$-th view. Consider the clustering problem on the dataset, each cluster is represented by a clustering center $\boldsymbol{\mu}_j^{(v)}, j = 1, ..., K$, where $K$ is the number of clusters. To achieve better clustering performance, we map the original feature space $\mathbf{X}^{(v)}$ to a latent embedded feature space $\mathbf{Z}^{(v)}$ through a non-linear mapping function

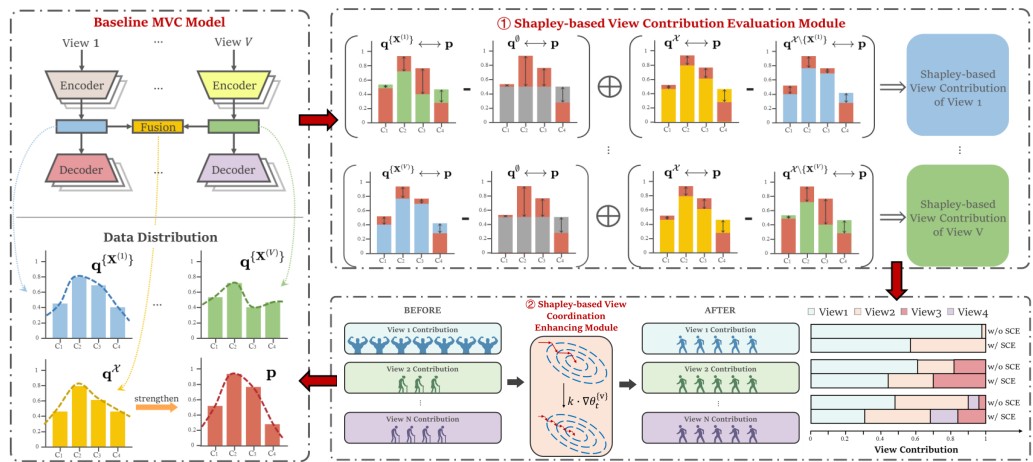

Figure 2: The framework of our proposed SCE model. SCE framework can be applied to most mainstream DMVC methods. After obtaining the cluster distribution from the selected method, SCE iteratively using the following two modules: View Contribution Evaluation Module and View Cooperation Enhancing Module. View Contribution Evaluation Module computes the marginal contribution of views in each combination(Shapley value), in which the optimal transport distance between the view cluster distribution and the strengthened cluster distribution after fusion serves as the utility function of Shapley formula. View Cooperation Enhancing Module controls the convergence speed ratio of different views through the calculated view contribution.

$f^{(v)}_{\theta^{(v)}} : \quad \mathbf{X}^{(v)} \to \quad \mathbf{Z}^{(v)}$, where $\theta^{(v)}$ represents the learnable parameters of mapping function for the $v$-th view.

## 3.2 View Contribution Evaluation Module

**Modified Shapley Value for Multi Views**

To assess the contributions of each view in MVC tasks, we treat each view as a player and introduce the Shapley value referring to Eq. (1). Let $\mathcal{X} = \{\mathbf{X}^{(i)}\}_{i=1}^{V}$ be the set of all views, $\mathcal{S}_v = \{\mathcal{S} \subseteq \mathcal{X} | \mathbf{X}^{(v)} \in \mathcal{S}\}$ represent the set of all subsets of $\mathcal{X}$ that include the $v$-th view $\mathbf{X}^{(v)}$. Define an internal metric of clustering $E$ independently of labels to serve as a utility function, which will be elaborated on in the next subsection. Then the Shapley value of the $v$-th view can be calculated as

$$Shapley_v = \sum_{s \in \mathcal{S}_v} \frac{(|s| - 1)!(V - |s|)!}{V!} \left[ E(s) - E(s \backslash \{\mathbf{X}^{(v)}\}) \right], \tag{2}$$

where $|s|$ denotes the number of views in the view-set $s$.

**Utility Function in Shapley Value**

In Shapley value, the utility function is used to evaluate the benefit created by a coalition. In DMVC, where each view acts as a participant, the utility function needs to be designed to reflect the contribution of any combination of views in fusion, without the need for label assistance. In this scenario, we constructed a novel function $E$ using the following method.

Inspired by the DEC model [50], we assume that the data from each view follows a Student's t-distribution. Following the approach outlined by [51], we utilize the t-distribution as a kernel function to evaluate the distance between the sample embeddings $\mathbf{z}_i$ and the cluster centers $\boldsymbol{\mu}_j$ within a view:

$$\mathbf{q}_{ij} = \frac{(1 + \|\mathbf{z}_i - \boldsymbol{\mu}_j\|^2/\alpha)^{-\frac{\alpha+1}{2}}}{\sum_{j'}(1 + \|\mathbf{z}_i - \boldsymbol{\mu}_{j'}\|^2/\alpha)^{-\frac{\alpha+1}{2}}}, \tag{3}$$

where $\alpha$ represents the degrees of freedom in the $t$-distribution. $\mathbf{q}_{ij}$ denotes the probability of assigning the feature $\mathbf{z}_i$ to the cluster center $\boldsymbol{\mu}_j$.

Fix the degrees of freedom $\alpha$ to 1, the cluster distribution of multi-view data $\mathbf{q}$ can be considered as a combination of single-view cluster distribution based on a weight matrix $\boldsymbol{\pi}$:

$$\mathbf{q}_{ij} = \frac{\sum_v \boldsymbol{\pi}_j^{(v)}(1 + \|\mathbf{z}_i^{(v)} - \boldsymbol{\mu}_j^{(v)}\|^2)^{-1}}{\sum_{j'} \sum_{v'} \boldsymbol{\pi}_{j'}^{(v')}(1 + \|\mathbf{z}_i^{(v')} - \boldsymbol{\mu}_{j'}^{(v')}\|^2)^{-1}}, \tag{4}$$

where the weight matrix $\boldsymbol{\pi} \in \mathbb{R}^{K \times V}$ represents the importance of the cluster centers $\boldsymbol{\mu}_j$ in different views. This weight matrix $\boldsymbol{\pi}$ can be obtained by performing row normalization on an unconstrained trainable matrix $\mathbf{W}$:

$$\boldsymbol{\pi}_j^{(v)} = \frac{e^{\mathbf{W}_j^{(v)}}}{\sum_{i=1}^{V} e^{\mathbf{W}_j^{(i)}}}. \tag{5}$$

For any given combination of an $m$-view union $\mathcal{U} = \{\mathbf{X}^{(u_1)}, \mathbf{X}^{(u_2)}, ..., \mathbf{X}^{(u_m)}\}$, we can obtain the weight matrix $\boldsymbol{\pi}^{\mathcal{U}}$ by selecting the corresponding columns in the matrix $\mathbf{W}$. By applying Eq. (3), we derive the data distribution for the fused view, which is denoted as $\mathbf{q}^{\mathcal{U}}$.

In particular, if $\mathcal{U} = \emptyset$, which means no view is involved in the clustering process, we consider $\mathbf{q}^{\emptyset}$ to be a uniform distribution. This means that the probability of any embedding $\mathbf{z}_i$ belonging to any cluster $\boldsymbol{\mu}_j$ is equal and given by $1/K$, where $K$ is the total cluster numbers. In this case, we denote the resulting data distribution $\mathbf{q}^{\emptyset}$ as

$$\mathbf{q}^{\emptyset} = \frac{1}{K} \mathbf{1}_{N \times K}, \tag{6}$$

where $\mathbf{1}_{N \times K} \in \mathbb{R}^{N \times K}$ is a matrix in which all elements are equal to 1.

According to the principle of self-distillation in DEC, we sharpen the fused distribution $\mathbf{q}^{\mathcal{X}}$ from all views to obtain a more powerful target distribution $\mathbf{p}$:

$$\mathbf{p}_{ij} = (\mathbf{q}_{ij}^{\mathcal{X}})^2 / \sum_j (\mathbf{q}_{ij}^{\mathcal{X}})^2. \tag{7}$$

Since we consider the distribution $\mathbf{p}$ as the trend of the fusion process, the closer the fusion distribution $\mathbf{q}^{\mathcal{U}}$ (combined with $\mathcal{U} = \{\mathbf{X}^{(u_1)}, \mathbf{X}^{(u_2)}, ..., \mathbf{X}^{(u_m)}\}$) is to the distribution $\mathbf{p}$, the higher the participation of view set $\mathcal{U}$ in the fusion process. Therefore, we can use the optimal transport distance from distribution $\mathbf{q}^{\mathcal{U}}$ to distribution $\mathbf{p}$ as a measure of confidence for distribution $\mathbf{q}^{\mathcal{U}}$. Additionally, based on our desire for the metric to possess monotonicity and boundedness, we define the utility function based on distribution transportation as:

$$E(\mathcal{U}) = \frac{1}{1 + OT(\mathbf{p}, \mathbf{q}^{\mathcal{U}})}, \tag{8}$$

where $OT(\cdot, \cdot)$ is calculated as wasserstein distance and $E(\cdot) \in (0, 1]$. The closer the data distribution is to the pseudo-labels $\mathbf{p}$, the larger the value of the metric $E(\cdot)$.

In the process of computing the optimal transport $OT(\cdot, \cdot)$, we consider the distribution $\mathbf{q}$ as a probability set where $K$ cluster centers are assigned by $N$ samples. The probability of each cluster center is $1/K$. For example, when calculating $OT(\mathbf{q}^{\{\mathbf{X}^{(1)}\}}, \mathbf{q}^{\{\mathbf{X}^{(2)}\}})$, the distribution of views can be represented as $[1/K, 1/K, ..., 1/K]^T$, and the distance matrix $\{\mathbf{D}_{ij}\}_{K \times K}$ can be calculated as

$$\mathbf{D}_{ij} = \| \boldsymbol{\xi}_i^{(1)} - \boldsymbol{\xi}_j^{(2)} \|_2^2, \tag{9}$$

where the $j$-th cluster center for the $v$-th view can be represented as $\boldsymbol{\xi}_j^{(v)} = \{\mathbf{q}_{ij}^{(v)}\}_{N \times 1}$.

**Shapley-based View Contribution: a Formal Representation**

In the above subsections, we have already discussed the integration of cooperative game theory with multi-view clustering tasks. We have also defined a utility function $E$ that reflects the accuracy of clustering. Now, combining the Eq. (2) and (8), we present the formal representation of view-contribution through Shapley values:

$$Shapley_v = \sum_{s \in \mathcal{S}_v} \frac{|s - 1|!(V - |s|)!}{V!} \left[ \frac{1}{1 + OT(\mathbf{p}, \mathbf{q}^s)} - \frac{1}{1 + OT(\mathbf{p}, \mathbf{q}^{s \setminus \{\mathbf{X}^{(v)}\}})} \right]. \tag{10}$$

The sum of all view contributions is then calculated as:

$$\sum_v Shapley_v = \frac{1}{1 + OT(\mathbf{p}, \mathbf{q}^{\mathcal{X}})} - \frac{1}{1 + OT(\mathbf{p}, \mathbf{q}^{\emptyset})}. \tag{11}$$

As $\mathbf{p}$ and $\mathbf{q}^{\mathcal{X}}$ evolve throughout training, the sum of Shapley values for all views also fluctuate accordingly. To obtain the contribution proportions of each view to the fused view, let

$$\varphi_v = \frac{Shapley_v}{\sum_{i=1}^{V} Shapley_i}, \tag{12}$$

where $\varphi_v$ is the normalized Shapley value, determining the respective contribution of different views.

### 3.3 Theoretical Analysis

To further demonstrate the generalization performance of the View Contribution Evaluation Module, we apply it to the alignment-based method and the joint method, respectively. The relevant theoretical derivations are given below.

**Theorem 1:** For alignment-based methods, considering the representation $\{\mathbf{z}_i^{(1)}\}_{i=1}^N$ and $\{\mathbf{z}_i^{(2)}\}_{i=1}^N$ on two views, with the infoNCE contrastive learning loss:

$$\ell_i^{(1)} = -\log \frac{\exp(\mathbf{z}_i^{(1)\top} \mathbf{z}_i^{(2)}/\tau_l)}{\sum_{j=1}^N \left[ \exp(\mathbf{z}_i^{(1)\top} \mathbf{z}_j^{(1)}/\tau_l) + \exp(\mathbf{z}_i^{(1)\top} \mathbf{z}_j^{(2)}/\tau_l) \right]}, \tag{13}$$

$$\ell_{con}(1,2) = \frac{1}{2N} \sum_{i=1}^N \left( \ell_i^{(1)} + \ell_i^{(2)} \right), \tag{14}$$

where $N$ is the sample number and $\tau > 0$ is a scalar temperature hyperparameter. $\ell_{con}(1,2)$ brings the representations between views closer, leads to the average of view contributions, *i.e.*, $|\phi_1 - \phi_2| \to 0$.

**Theorem 2:** For joint methods, considering the representation $\mathbf{Z}^{(1)} = f_{\theta^{(1)}}^{(1)}(\mathbf{X}^{(1)}; \theta^{(1)})$ and $\mathbf{Z}^{(2)} = f_{\theta^{(2)}}^{(2)}(\mathbf{X}^{(2)}; \theta^{(2)})$ on two views, with a simple view-level weight fusion during the fusion process: $\mathbf{Z} = w_1 \mathbf{Z}^{(1)} + w_2 \mathbf{Z}^{(2)}$. There are situations where view 2 is dominated by view 1 and cannot make its contribution, *i.e.*, $w_1 \gg w_2$.

**Remark:** Theorem 1 indicates that alignment-based methods tend to align the contributions of different views. Under our assumption, the cooperation between views is already sufficient, optimizing the model based on contributions can only lead to limited improvements. Theorem 2 suggests that in joint methods, one view may suppress another, highlighting insufficient cooperation between views. In such cases, enhancing cooperation between views theoretically improves model performance. We will elaborate on the View Cooperation Enhancing Module in Section 3.4 and validate these conclusions through experiments. The proofs for Theorems 1 and 2 can be found in Appendix.

### 3.4 View Cooperation Enhancing Module

After acquiring contributions of different views, we seek to enhance their cooperation by dynamically regulating views' training speeds according to these contributions. This proportional control guarantees active involvement of all views in the integration process. For this purpose, we introduce the convergence speed ratio, labeled as $k$, derived from the contributions of each view.

To ensure the applicability of our **S**hapley-based **C**ooperation **E**nhancing(**SCE**) method to most multi-view clustering tasks, we consider a typical multi-view clustering framework: firstly, obtain embeddings for each view by utilizing autoencoders; then, fuse the embeddings from different views using a global optimization goal. Let $f_{\theta^{(v)}}^{(v)}(\mathbf{X}^{(v)}; \theta^{(v)})$ represent the encoder model for the $v$-th view, $g$ denote the fusion method for multiple views, and $l$ represent the loss function for computing the fusion loss. The overall fusion loss $L$ can be represented as

$$L(\theta_t) = l(g(f_{\theta^{(1)}}^{(1)}(\mathbf{X}^{(1)}; \theta_t^{(1)}), f_{\theta^{(2)}}^{(2)}(\mathbf{X}^{(2)}; \theta_t^{(2)}), ..., f_{\theta^{(V)}}^{(V)}(\mathbf{X}^{(V)}; \theta_t^{(V)}))). \tag{15}$$

In practice, the parameters for the $v$-th view is updated as

$$\theta_{t+1}^{(v)} = \theta_t^{(v)} - \eta \nabla_{\theta^{(v)}} L(\theta_t^{(v)}). \tag{16}$$

With the normalized Shapley values calculated by Eq. (12) to dynamically monitor the contribution differences between different views, we are able to adaptively modulate the gradients via an empirical formula:

$$k_v = e^{\tau(\varphi_{min}/\varphi_v - 1)}, \tag{17}$$

where $\tau$ is a temperature hyperparameter to control the degree of modulation. Thus $k_v$ is not greater than 1, and the larger the contribution of the view, the smaller the $k_v$. We integrate the coefficient $k_i$ into the Adam optimization method, and the update of $\theta_{t+1}^{(v)}$ at iteration $t$ is as follows:

$$\theta_{t+1}^{(v)} = \theta_t^{(v)} - k_v \cdot \eta \frac{\partial l}{\partial g} \cdot \frac{\partial g}{\partial f_v} \cdot \frac{\partial f_v}{\partial \theta_t^{(v)}}, \tag{18}$$

where $k_v$ is used as a relative convergence speed ratio for the $v$-th view in gradient descent.

**Theorem 3:** For joint methods that use view-level weight fusion $\mathbf{Z} = w_1 \mathbf{Z}^{(1)} + w_2 \mathbf{Z}^{(2)}$, where $\mathbf{Z}^{(1)} = f_{\theta^{(1)}}^{(1)}(\mathbf{X}^{(1)}; \theta^{(1)})$ and $\mathbf{Z}^{(2)} = f_{\theta^{(2)}}^{(2)}(\mathbf{X}^{(2)}; \theta^{(2)})$, gradient modulation in Eq. (18) allows the two views to contribute more evenly, *i.e.*, $\frac{w_1}{w_2} \to 1$.

**Remark:** Theorem 3 indicates that through dynamic gradient adjustments, we can harmonize the learning progress among different views, attain a more average distribution of contributions, and enhance the cooperation among views. The proof for Theorem 3 can be found in Appendix.

---

**Algorithm 1: S**hapley-based **C**ooperation **E**nhancing **M**ulti-view **C**lustering(**SCE-MVC**)

---

**Input:** Multi-view dataset $\{\mathbf{X}^{(v)}\}_{v=1}^V$, temperature hyperparameter $\tau$
**Output:** cluster labels $\hat{y}$

```
// Warm-up training
```
Obtain initial cluster assignments through a selected algorithm;
```
// Optimization with SCE module
```
**while** *the fusion loss has not converge* **do**
    ```// View Contribution Evaluation```
    Obtain shapley value for each view with Eq. (4), (7) and (10);
    Normalize the shapley value as view contribution with Eq. (12);
    ```// View Cooperation Enhancing```
    Calculate the convergence speed ratio $k_v$ for each view with Eq. (17);
    Update the parameters of each view relatively with Eq. (18);
return;

---

### 3.5 Implementation

This subsection outlines the implementation of our SCE module, detailed in Algorithm 1. After obtaining initial cluster center assignments, we iteratively employ the following two modules:

**View Contribution Evaluation**. First, we calculate the cluster distribution for each individual view as well as the combined view. Next, we compute an enhanced distribution $\mathbf{p}$. After utilizing the optimal transport distance between distributions as the utility function, we can calculate view contributions.

**View Cooperation Enhancing**. We utilize the calculated contribution values to influence the training process of each view using a convergence ratio $k$. The larger the contribution value of a view, the less the convergence ratio $k$ will be. This will coordinate the training process of the views, enabling better cooperation among views, and allowing them to collaborate more effectively.

## 4 Experiments

In this section, we implement experiments on alignment-based method and joint method to verify the effectiveness of the proposed theorems and SCE module by addressing the following questions:

**RQ1**: For alignment-based MVC methods, do the contributions of views calculated by the View Contribution Evaluation Module exhibit relative uniformity, thereby validating **Theorem 1**?

**RQ2**: In the case of uniform view contributions, can model performance be further enhanced by the View Cooperation Enhancing Module?

**RQ3**: For joint methods, are extreme view contributions present, thereby validating **Theorem 2**?

**RQ4**: Can model performance be enhanced by harmonizing view contributions through View Cooperation Enhancing Module, thereby validating **Theorem 3**?

**RQ5**: How do the hyper-parameter $\tau$ impact the performance of SCE-MVC?

Table 1: Dataset summary.

| Dataset | Views | Samples | Clusters | Dataset | Views | Samples | Clusters |
|---------|-------|---------|----------|---------|-------|---------|----------|
| CUB | 2 | 600 | 10 | UCI-digit | 3 | 2000 | 10 |
| Caltech101-7 | 2 | 1474 | 7 | STL10 | 4 | 13000 | 10 |
| HandWritten | 3 | 2000 | 10 | Reuters | 5 | 1200 | 6 |

Table 2: Analyzing view contribution under InfoNCE+Kmeans and ProImp frameworks on CUB and Caltech101-7 datasets.

| Dataset | Method | View Contribution | | Metrics | | |
|---------|--------|---------|---------|---------|---------|---------|
| | | $\phi_1$ | $\phi_2$ | ACC | NMI | ARI |
| CUB | InfoNCE+Kmeans | 0.491 | 0.509 | 0.715 | **0.748** | **0.626** |
| | InfoNCE+Kmeans+**SCE** | 0.493 | 0.507 | **0.717** | 0.747 | **0.626** |
| | ProIMP[43] | 0.556 | 0.443 | 0.825 | 0.756 | 0.671 |
| | ProIMP+**SCE** | 0.484 | 0.516 | **0.832** | **0.762** | **0.678** |
| Caltech101-7 | InfoNCE+Kmeans | 0.484 | 0.516 | 0.351 | **0.486** | 0.272 |
| | InfoNCE+Kmeans+**SCE** | 0.496 | 0.504 | **0.364** | 0.485 | **0.281** |
| | ProIMP[43] | 0.489 | 0.511 | 0.382 | 0.468 | **0.281** |
| | ProIMP+**SCE** | 0.499 | 0.501 | **0.382** | **0.470** | 0.279 |

## 4.1 Datasets, Metrics and Experimental settings

Our experiments utilized six multi-view datasets, including CUB[2], Caltech101-7[3], HandWritten[4], UCI-digit[5], STL10[6] and Reuters[7]. The detailed information of these datasets is listed in the Table 1. Meanwhile, three widely used metrics are adopted in our experiment, including clustering accuracy (ACC), normalized mutual information (NMI) and adjusted rand index(ARI).

For fairness, We conduct all experiments on PyTorch platform using the NVIDIA 2060 GPU. Besides, ten state-of-the-art MVC methods are introduced: DEMVC[13], CoMVC[14], SiMVC[14], SDSNE[15], MFLVC[16], SDMVC[17], DSMVC[18], APADC[52], DMJC[35] and ProImp[43].

## 4.2 Evaluate on Alignment-based Methods(RQ1 & RQ2)

In this section, we apply the SCE methods to two alignment-based MVC methods:

- Utilizing infoNCE as a contrastive loss and employing Kmeans after concatenation.

- Incorporating contrastive learning at both the sample and prototype levels as ProIMP[43].

We applied these two approaches to the CUB and Caltech101-7 datasets, each comprising two views. Detailed experimental results are presented in the Table 2. The results lead to following conclusions:

---

[2] http://www.vision.caltech.edu/visipedia/CUB-200.html

[3] https://data.caltech.edu/records/mzrjq-6wc02

[4] https://archive.ics.uci.edu/ml/datasets/Multiple+Features

[5] https://cs.nyu.edu/âĹijroweis/data.html

[6] https://cs.stanford.edu/~acoates/stl10/

[7] http://archive.ics.uci.edu/ml/datasets/Reuters-21578+Text+Categorization+Collection

Table 3: Multi-view clustering performance on six benchmark datasets. The optimal results are marked in bold, and the suboptimal values are underlined. O/M denotes out-of-memory error encountered during the training process.

| Methods | Caltech101-7 | | | CUB | | | UCI-digit | | | HandWritten | | | STL10 | | | Reuters | | |
|---|---|---|---|---|---|---|---|---|---|---|---|---|---|---|---|---|---|---|
| Metrics | ACC | NMI | ARI | ACC | NMI | ARI | ACC | NMI | ARI | ACC | NMI | ARI | ACC | NMI | ARI | ACC | NMI | ARI |
| DEMVC[13] | 0.352 | 0.289 | 0.260 | 0.520 | 0.538 | 0.377 | 0.623 | 0.605 | 0.479 | 0.275 | 0.294 | 0.221 | 0.300 | 0.253 | 0.077 | 0.450 | 0.211 | 0.221 |
| CoMVC[14] | 0.530 | 0.375 | 0.152 | 0.358 | 0.503 | 0.278 | 0.464 | 0.494 | 0.455 | 0.369 | 0.355 | 0.387 | 0.236 | 0.163 | 0.056 | 0.307 | 0.073 | 0.189 |
| SiMVC[14] | 0.523 | 0.268 | 0.136 | 0.323 | 0.475 | 0.252 | 0.208 | 0.166 | 0.358 | 0.308 | 0.214 | 0.359 | 0.160 | 0.063 | 0.040 | 0.336 | 0.104 | 0.192 |
| SDSNE[15] | 0.568 | 0.508 | 0.361 | 0.773 | **0.787** | **0.670** | 0.846 | **0.891** | 0.816 | 0.876 | 0.878 | 0.819 | O/M | O/M | O/M | 0.233 | 0.203 | 0.017 |
| MFLVC[16] | 0.401 | 0.424 | 0.261 | 0.668 | 0.632 | 0.489 | 0.920 | 0.854 | 0.740 | 0.862 | 0.855 | 0.757 | 0.311 | 0.254 | 0.157 | 0.399 | 0.200 | 0.207 |
| SDMVC[17] | 0.366 | 0.335 | 0.237 | 0.692 | 0.654 | 0.517 | 0.630 | 0.643 | 0.601 | 0.495 | 0.519 | 0.390 | 0.283 | **0.262** | 0.136 | 0.177 | 0.047 | 0.113 |
| DSMVC[18] | 0.454 | 0.423 | 0.292 | 0.273 | 0.170 | 0.100 | 0.855 | 0.807 | 0.732 | 0.911 | 0.840 | 0.813 | 0.275 | 0.193 | 0.102 | 0.438 | 0.181 | 0.120 |
| APADC[52] | 0.562 | 0.422 | 0.303 | 0.672 | 0.692 | 0.552 | 0.629 | 0.686 | 0.549 | 0.711 | 0.703 | 0.563 | 0.283 | 0.196 | 0.078 | 0.245 | 0.027 | 0.037 |
| DMJC[35] | 0.469 | 0.411 | 0.309 | 0.758 | 0.749 | 0.617 | 0.871 | 0.825 | 0.767 | 0.893 | 0.846 | 0.805 | 0.305 | 0.241 | 0.155 | 0.485 | 0.293 | **0.241** |
| DMJC+**SCE** | **0.583** | **0.513** | **0.457** | **0.797** | 0.770 | 0.655 | **0.927** | 0.865 | **0.847** | **0.938** | **0.881** | **0.870** | **0.330** | 0.256 | **0.162** | **0.533** | **0.294** | 0.237 |

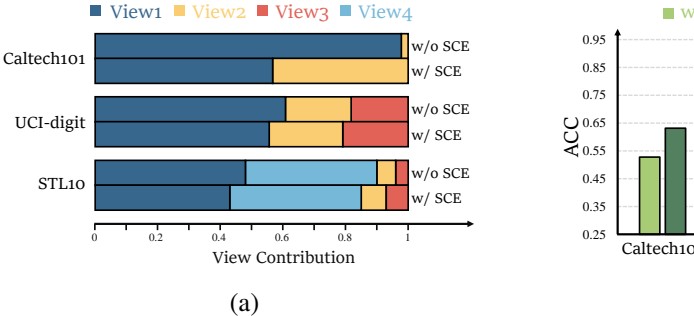

Figure 3: (a) View contribution is more balanced with SCE. (b) Fusion performs better with SCE.

• Assessed with the View Contribution Evaluation Module, the contributions of the two views in alignment-based methods are remarkably similar, validating the assertion of Theorem 1.

• As the contributions of the views tend to equalize, the utilization of the View Cooperation Enhancing Module approach only yields subtle improvements to the alignment-based methods.

## 4.3 Evaluate then Cooperate on Joint Methods(RQ3 & RQ4)

In this subsection, we apply the SCE model to a joint MVC methods(DMJC[35]). We have carried out comprehensive experiments on six datasets with varying numbers of views, as depicted in Table 3 and Fig. 3. In Table 3, we showcase the performance of SCE-MVC in comparison to ten benchmark methods. Fig. 3 illustrates the ablation study of DMJC with and without SCE. Based on the aforementioned experimental results, we draw the following conclusions:

• Fig. 3(a) illustrates the changes in view contributions with and without the SCE module across three datasets. Prior to the use of the SCE module, there is a scenario where certain views dominate while others are suppressed, a pattern particularly evident in the Caltech101-7 dataset, aligning well with the conclusions of Theorem 2.

• After introducing the View Contribution Enhancing Module, the contributions of the views become more balanced, leading to a significant improvement in the model's performance. Specifically, the ACC, NMI, and ARI metrics on the Caltech101-7 dataset all witness an increase of over 10 percentage points, thus validating the conclusion of Theorem 3. More detailed contribution comparison data can be found in the Appendix.

Table 4: Sensitive analysis on UCI-digit and STL10 datasets. The optimal results are marked in bold.

| $\tau$ | 0.5 | | | 1.0 | | | 1.5 | | |
|---|---|---|---|---|---|---|---|---|---|
| **Datasets** | **ACC** | **NMI** | **ARI** | **ACC** | **NMI** | **ARI** | **ACC** | **NMI** | **ARI** |
| UCI-digit | 0.871 | 0.825 | 0.767 | 0.882 | 0.830 | 0.777 | 0.886 | 0.834 | 0.788 |
| STL10 | 0.303 | 0.257 | 0.152 | **0.330** | 0.256 | **0.162** | 0.320 | 0.259 | 0.159 |
| $\tau$ | 2.0 | | | 2.5 | | | 3.0 | | |
| **Datasets** | **ACC** | **NMI** | **ARI** | **ACC** | **NMI** | **ARI** | **ACC** | **NMI** | **ARI** |
| UCI-digit | 0.903 | 0.842 | 0.805 | 0.927 | **0.865** | **0.847** | **0.930** | 0.860 | 0.842 |
| STL10 | 0.317 | **0.263** | 0.160 | 0.319 | 0.260 | 0.160 | 0.317 | 0.259 | 0.160 |

## 4.4 Sensitive Analysis(RQ5)

In SCE-MVC, the singular hyperparameter $\tau$ is essential for fine-tuning the convergence equilibrium among distinct views according to their contributions. In sensitive analysis, we varied $\tau$ within $\{0.5, 1, 1.5, 2, 2.5, 3\}$, as illustrated in Table 4. As $\tau$ increases, the changes in clustering results metrics become less pronounced, which can be attributed to the characteristics of the exponential functions in Eq. (17). Specifically, as $\tau$ increases, the convergence ratio $k_v$ tends to be closer to zero.

## 5 Conclusion

This paper explores the roles of views in multi-view clustering through the lens of game theory. We introduce a novel contribution evaluation module founded on the Shapley value. Our experiments showcase the adaptability of this module across mainstream MVC frameworks, accurately assessing each view's impact. By scrutinizing the functions of diverse views during fusion, we devise a cooperation enhancing module to bolster cooperation among views. Extensive dataset validations underscore the effectiveness and robust applicability of our approach and theorems.

## Acknowledgments and Disclosure of Funding

This work is supported by National Natural Science Foundation of China under Grant No. 62276271, 62325604, 62406329, 62476280, 62476281 and National Science and Technology Innovation 2030 Major Project under Grant No. 2022ZD0209103.

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

# A Proof of Theorems

## A.1 Proof of Theorem 1

**Theorem 1:** For alignment-based methods, considering the representation $\{\mathbf{z}_i^{(1)}\}_{i=1}^N$ and $\{\mathbf{z}_i^{(2)}\}_{i=1}^N$ on two views, with the infoNCE contrastive learning loss:

$$\ell_i^{(1)} = -\log \frac{\exp(\mathbf{z}_i^{(1)^\top} \mathbf{z}_i^{(2)}/\tau_l)}{\sum_{j=1}^N \left[\exp(\mathbf{z}_i^{(1)^\top} \mathbf{z}_j^{(1)}/\tau_l) + \exp(\mathbf{z}_i^{(1)^\top} \mathbf{z}_j^{(2)}/\tau_l)\right]}, \tag{19}$$

$$\ell_{con}(1,2) = \frac{1}{2N} \sum_{i=1}^N \left(\ell_i^{(1)} + \ell_i^{(2)}\right), \tag{20}$$

where $N$ is the sample number and $\tau > 0$ is a scalar temperature hyperparameter. $\ell_{con}(1,2)$ brings the representations between views closer, leads to the average of view contributions, *i.e.*, $|\phi_1 - \phi_2| \to 0$.

**Proof:** Referring to Definition of **Perfect Alignment** in [53], we say the two views are perfectly aligned if $\mathbf{z}_i^{(1)} = \mathbf{z}_i^{(2)}, i = 1, 2, ..., N$. Assuming that with the training of contrastive learning, the representations of the two views tend to align perfectly. Then, there exists a limiting representation $Z^*$ such that $\lim \|Z_1 - Z^*\| = \lim \|Z_2 - Z^*\| = 0$.

Due to the convergence of sample representations from both views, the centroids and sample distributions of the views also tend to converge, *i.e.* $\lim \|\boldsymbol{\mu}^{(1)} - \boldsymbol{\mu}^*\| = \lim \|\boldsymbol{\mu}^{(2)} - \boldsymbol{\mu}^*\| = 0$ and $\lim \|\mathbf{q}^{\{1\}} - \mathbf{q}^*\| = \lim \|\mathbf{q}^{\{2\}} - \mathbf{q}^*\| = 0$, where $\boldsymbol{\mu}^*$ and $\mathbf{q}^*$ represent the clustering centroids and sample distributions respectively when the two views are perfectly aligned.

Therefore, the utility function based on distribution transportation tends to converge towards a same value, namely

$$E(\{i\}) = \frac{1}{1 + OT(\mathbf{p}, \mathbf{q}^*)}, i \in \{1, 2\}. \tag{21}$$

According to the definition of Shapley values, the same utility function values yield the same Shapley values. Therefore, the normalized Shapley value $|\phi_1 - \phi_2| \to 0$.

## A.2 Proof of Theorem 2

**Theorem 2:** For joint methods, considering the representation $\mathbf{Z}^{(1)} = f_{\theta^{(1)}}^{(1)}(\mathbf{X}^{(1)}; \theta^{(1)})$ and $\mathbf{Z}^{(2)} = f_{\theta^{(2)}}^{(2)}(\mathbf{X}^{(2)}; \theta^{(2)})$ on two views, with a simple view-level weight fusion during the fusion process: $\mathbf{Z} = w_1 \mathbf{Z}^{(1)} + w_2 \mathbf{Z}^{(2)}$. There are situations where view 2 is dominated by view 1 and cannot make its contribution, *i.e.*, $w_1 \gg w_2$.

**Proof:** Suppose the representation of the two views:

$$\begin{cases} \mathbf{Z}^{(1)} = f_{\theta^{(1)}}^{(1)}(\mathbf{X}^{(1)}; \theta^{(1)}), \\ \mathbf{Z}^{(2)} = f_{\theta^{(2)}}^{(2)}(\mathbf{X}^{(2)}; \theta^{(2)}), \end{cases} \tag{22}$$

and view-level weight fusion is adopted in the fusion process:

$$\mathbf{Z} = w_1 \mathbf{Z}^{(1)} + w_2 \mathbf{Z}^{(2)}, \tag{23}$$

where $w_1$ and $w_2$ represent the weights of View 1 and View 2 respectively, satisfying $w_1 + w_2 = 1$.

Conduct Kmeans on the fused representation $\mathbf{Z}$, the overall optimization goal is

$$E = \frac{1}{2} \sum_{i=1}^K \sum_{\mathbf{z} \in \mathcal{C}_i} \| \mathbf{z} - \boldsymbol{\mu}_i \|_2^2, \tag{24}$$

where $K$ is the number of clusters, $\mathcal{C}_i$ is the sample set of the $i$-th cluster, and $\boldsymbol{\mu}_i$ is the centroid of $\mathcal{C}_i$, which can be represented as

$$\boldsymbol{\mu}_i = \frac{1}{|\mathcal{C}_i|} \sum_{\mathbf{z} \in \mathcal{C}_i} \mathbf{z}, \tag{25}$$

where $|\mathcal{C}_i|$ represent the number of samples in the sample set $\mathcal{C}_i$.

For arbitrary sample $\mathbf{z}_j \in \mathcal{C}_{k_j}$, the corresponding centroid is

$$\boldsymbol{\mu}_{k_j} = w_1 \boldsymbol{\mu}_{k_j}^{(1)} + w_2 \boldsymbol{\mu}_{k_j}^{(2)}. \tag{26}$$

The loss corresponding to the sample $\mathbf{z}_j$ is

$$\begin{aligned}
E_j &= \frac{1}{2} \| \mathbf{z}_j - \boldsymbol{\mu}_{k_j} \|_2^2 \\
&= \frac{1}{2} \| \left( w_1 \mathbf{z}_j^{(1)} + w_2 \mathbf{z}_j^{(2)} \right) - \left( w_1 \boldsymbol{\mu}_{k_j}^{(1)} + w_2 \boldsymbol{\mu}_{k_j}^{(2)} \right) \|_2^2 \\
&= \frac{1}{2} \| w_1 \left( \mathbf{z}_j^{(1)} - \boldsymbol{\mu}_{k_j}^{(1)} \right) + w_2 \left( \mathbf{z}_j^{(2)} - \boldsymbol{\mu}_{k_j}^{(2)} \right) \|_2^2 \\
&= \frac{1}{2} \| w_1 \left( \mathbf{z}_j^{(1)} - \boldsymbol{\mu}_{k_j}^{(1)} \right) + (1 - w_1) \left( \mathbf{z}_j^{(2)} - \boldsymbol{\mu}_{k_j}^{(2)} \right) \|_2^2.
\end{aligned} \tag{27}$$

Derive with respect to $w_1$ for $E_j$,

$$\begin{aligned}
\frac{\partial E_j}{\partial w_1} &= \left( \left( \mathbf{z}^{(1)} - \boldsymbol{\mu}_{k_j}^{(1)} \right) - \left( \mathbf{z}_j^{(2)} - \boldsymbol{\mu}_{k_j}^{(2)} \right) \right)^T \left( w_1 \left( \mathbf{z}_j^{(1)} - \boldsymbol{\mu}_{k_j}^{(1)} \right) + w_2 \left( \mathbf{z}_j^{(2)} - \boldsymbol{\mu}_{k_j}^{(2)} \right) \right) \\
&= w_1 \| \mathbf{z}_j^{(1)} - \boldsymbol{\mu}_{k_j}^{(1)} \|_2^2 - w_2 \| \mathbf{z}_j^{(2)} - \boldsymbol{\mu}_{k_j}^{(2)} \|_2^2 - (w_1 - w_2) \left( \mathbf{z}^{(1)} - \boldsymbol{\mu}_{k_j}^{(1)} \right)^T \left( \mathbf{z}_j^{(2)} - \boldsymbol{\mu}_{k_j}^{(2)} \right).
\end{aligned} \tag{28}$$

The above only considered the individual sample $\mathbf{z}_j$ to the total loss. When considering all samples

$$E = \sum_{k=1}^{K} \sum_{j=1}^{|\mathcal{C}_k|} E_j, \tag{29}$$

then derive with respect to $w_1$ for $E$,

$$\frac{\partial E}{\partial w_1} = \sum_{k=1}^{K} \sum_{j=1}^{|\mathcal{C}_k|} \frac{\partial E_j}{\partial w_1}. \tag{30}$$

If $N$ represents the total number of samples, taking the expectation $\text{Avg}(\cdot)$ of both sides of the equation yields

$$\begin{aligned}
\text{Avg} \left( \frac{\partial E}{\partial w_1} \right) &= \sum_{k=1}^{K} \sum_{j=1}^{|\mathcal{C}_k|} \text{Avg} \left( \frac{\partial E_j}{\partial w_1} \right) \\
&= N \cdot \text{Avg} \left( \frac{\partial E_j}{\partial w_1} \right) \\
&= N \left( w_1 \left( \text{Avg} \left( \| \mathbf{z}_j^{(1)} - \boldsymbol{\mu}_{k_j}^{(1)} \|_2^2 \right) - \text{Avg} \left( \left( \mathbf{z}^{(1)} - \boldsymbol{\mu}_{k_j}^{(1)} \right)^T \left( \mathbf{z}_j^{(2)} - \boldsymbol{\mu}_{k_j}^{(2)} \right) \right) \right) \right. \\
&\quad \left. - w_2 \left( \text{Avg} \left( \| \mathbf{z}_j^{(2)} - \boldsymbol{\mu}_{k_j}^{(2)} \|_2^2 \right) - \text{Avg} \left( \left( \mathbf{z}^{(1)} - \boldsymbol{\mu}_{k_j}^{(1)} \right)^T \left( \mathbf{z}_j^{(2)} - \boldsymbol{\mu}_{k_j}^{(2)} \right) \right) \right) \right).
\end{aligned} \tag{31}$$

When $\frac{\partial E}{\partial w_1} = 0$, the analytical solution for $w_1$ can be calculated, and at this point

$$\frac{w_1}{w_2} = \frac{\| \mathbf{z}_j^{(2)} - \boldsymbol{\mu}_{k_j}^{(2)} \|_2^2 - \left( \mathbf{z}^{(1)} - \boldsymbol{\mu}_{k_j}^{(1)} \right)^T \left( \mathbf{z}_j^{(2)} - \boldsymbol{\mu}_{k_j}^{(2)} \right)}{\| \mathbf{z}_j^{(1)} - \boldsymbol{\mu}_{k_j}^{(1)} \|_2^2 - \left( \mathbf{z}^{(1)} - \boldsymbol{\mu}_{k_j}^{(1)} \right)^T \left( \mathbf{z}_j^{(2)} - \boldsymbol{\mu}_{k_j}^{(2)} \right)}, \tag{32}$$

where expression $\text{Avg}(\| \mathbf{z}_j^{(i)} - \boldsymbol{\mu}_{k_j}^{(i)} \|_2^2)$ reflects the level of difficulty in clustering for view $i$ ($i \in \{1,2\}$); the clearer the clustering structure, the smaller the $\text{Avg}(\| \mathbf{z}_j^{(i)} - \boldsymbol{\mu}_{k_j}^{(i)} \|_2^2)$.

And $\left(\mathbf{z}^{(1)} - \boldsymbol{\mu}_{k_j}^{(1)}\right)^T \left(\mathbf{z}_j^{(2)} - \boldsymbol{\mu}_{k_j}^{(2)}\right)$ can be seen as a dot product similarity, reflecting the correlation between the two views; the smaller the correlation between views, the closer $\left(\mathbf{z}^{(1)} - \boldsymbol{\mu}_{k_j}^{(1)}\right)^T \left(\mathbf{z}_j^{(2)} - \boldsymbol{\mu}_{k_j}^{(2)}\right)$ is to 0.

Assuming significant differences in clustering structures between two views, with view 1 being the dominant view over view 2, where $\text{Avg}(\|\mathbf{z}_j^{(1)} - \boldsymbol{\mu}_{k_j}^{(1)}\|_2^2) \ll \text{Avg}(\|\mathbf{z}_j^{(2)} - \boldsymbol{\mu}_{k_j}^{(2)}\|_2^2)$ and $\text{Avg}(\left(\mathbf{z}^{(1)} - \boldsymbol{\mu}_{k_j}^{(1)}\right)^T \left(\mathbf{z}_j^{(2)} - \boldsymbol{\mu}_{k_j}^{(2)}\right)) \approx 0$, then when the loss of multi-view clustering converges, the weights of the two views

$$\frac{w_1}{w_2} \approx \frac{\text{Avg}\|\mathbf{z}_j^{(2)} - \boldsymbol{\mu}_{k_j}^{(2)}\|_2^2}{\text{Avg}\|\mathbf{z}_j^{(1)} - \boldsymbol{\mu}_{k_j}^{(1)}\|_2^2} \gg 1, \tag{33}$$

which means in the fusion of views, view 1 takes the lead, while view 2 can only play a limited role. However, in unsupervised tasks, the clarity of the clustering structure is unrelated to the quality of the clustering results. Even though the clustering structure of view 2 may not be clear, the clustering results of view 2 could still be accurate. In this scenario, $w_1 \gg w_2$ can be considered that the cooperation between the two views is suboptimal.

### A.3 Proof of Theorem 3

**Theorem 3:** For joint methods that use view-level weight fusion $\mathbf{Z} = w_1 \mathbf{Z}^{(1)} + w_2 \mathbf{Z}^{(2)}$, where $\mathbf{Z}^{(1)} = f_{\theta^{(1)}}^{(1)}(\mathbf{X}^{(1)}; \theta^{(1)})$ and $\mathbf{Z}^{(2)} = f_{\theta^{(2)}}^{(2)}(\mathbf{X}^{(2)}; \theta^{(2)})$, gradient modulation in Eq. (18) allows the two views to contribute more evenly, *i.e.*, $\frac{w_1}{w_2} \to 1$.

**Proof:** Introducing the suppression factor $k$ (where $k < 1$) suppresses the convergence speed of the dominant view's parameters. In this case, the parameter updates become:

$$\begin{cases} \theta^{(1)} = \theta^{(1)} - k \cdot \eta \frac{\partial E}{\partial \theta^{(1)}} \\ \theta^{(2)} = \theta^{(2)} - \eta \frac{\partial E}{\partial \theta^{(2)}} \end{cases} \tag{34}$$

where the overall optimization objective for multi-view learning is given by

$$E = \frac{1}{2} \sum_{i=1}^K \sum_{\mathbf{z} \in \mathcal{C}_i} \|\mathbf{z} - \boldsymbol{\mu}_i\|_2^2. \tag{35}$$

Due to the equivalence $\|\mathbf{z} - \boldsymbol{\mu}_i\|_2^2 = \left\| w_1 \left(\mathbf{z}_j^{(1)} - \boldsymbol{\mu}_{k_j}^{(1)}\right) + w_2 \left(\mathbf{z}_j^{(2)} - \boldsymbol{\mu}_{k_j}^{(2)}\right) \right\|_2^2$, in accordance with the Cauchy-Schwarz inequality, we obtain:

$$\|\mathbf{z} - \boldsymbol{\mu}_i\|_2^2 \le \left(w_1^2 + w_2^2\right) \left( \left\|\mathbf{z}_j^{(1)} - \boldsymbol{\mu}_{k_j}^{(1)}\right\|_2^2 + \left\|\mathbf{z}_j^{(2)} - \boldsymbol{\mu}_{k_j}^{(2)}\right\|_2^2 \right). \tag{36}$$

Therefore, the upper bound on the overall loss $E$ is jointly determined by the individual view losses and the view weights. As both $\|\mathbf{z}_j^{(1)} - \boldsymbol{\mu}_{k_j}^{(1)}\|_2^2$ and $\|\mathbf{z}_j^{(2)} - \boldsymbol{\mu}_{k_j}^{(2)}\|_2^2$ can be seen as the Kmeans loss for a single view, during one iteration, if we suppress the gradient descent for the dominant view (view 1), $\|\mathbf{z}_j^{(1)} - \boldsymbol{\mu}_{k_j}^{(1)}\|_2^2$ would increase compared to not suppressing it, while $\|\mathbf{z}_j^{(2)} - \boldsymbol{\mu}_{k_j}^{(2)}\|_2^2$ remains unaffected. The upper bound on the overall optimization objective shifts, leading to differences in the single-view parameters and weights at convergence.

As the loss converges, the ratio of weights between the two views, $\frac{w_1}{w_2} \to 1$. This implies that the weight of the non-dominant view increases, allowing it to more fully engage in the fusion process. Consequently, there is improved cooperation between the views, enhancing the fusion process.

# B    Further Experiments

## B.1    View Contribution Comparison w/o SCE

In this subsection, we conducted additional ablation experiments on the SCE module based on the DMJC method across six datasets. Table 5 illustrates the variations in view contributions before and after using SCE. The experimental findings further corroborate our conclusions:

• There are cases where certain views in some datasets do not effectively contribute, such as the 2-nd view of Caltech101-7, the 2-nd view of CUB, the 3-rd and 4-th views of STL10, and the 5-th view of Reuters, all with contributions below 0.1.

• Through the SCE method, the contributions of suppressed views experienced significant growth, with the contribution of the 2-nd view of Caltech101-7 increasing from 0.032 to 0.433, and the contribution of the 5-th view of Reuters increasing from 0.107 to 0.232.

Table 5: View Contribution Comparison w/o SCE

| Dataset | Method | $\phi_1$ | $\phi_2$ | $\phi_3$ | $\phi_4$ | $\phi_5$ |
|---------|--------|------|------|------|------|------|
| Caltech101-7 | DMJC | 0.968 | 0.032 | \ | \ | \ |
| | DMJC+**SCE** | 0.567 | 0.433 | \ | \ | \ |
| CUB | DMJC | 0.974 | 0.026 | \ | \ | \ |
| | DMJC+**SCE** | 0.892 | 0.108 | \ | \ | \ |
| UCI-digit | DMJC | 0.671 | 0.191 | 0.138 | \ | \ |
| | DMJC+**SCE** | 0.636 | 0.197 | 0.167 | \ | \ |
| HandWritten | DMJC | 0.661 | 0.192 | 0.147 | \ | \ |
| | DMJC+**SCE** | 0.645 | 0.178 | 0.177 | \ | \ |
| STL10 | DMJC | 0.479 | 0.425 | 0.062 | 0.044 | \ |
| | DMJC+**SCE** | 0.432 | 0.409 | 0.083 | 0.076 | \ |
| Reuters | DMJC | 0.301 | 0.248 | 0.23 | 0.114 | 0.107 |
| | DMJC+**SCE** | 0.177 | 0.235 | 0.138 | 0.218 | 0.232 |

Table 6: Three views training on the UCI-digit dataset and fuse the results of different views. The optimal results are marked in bold, and the suboptimal values are underlined.

| Fusion Views | | | ACC | |
|-------|-------|-------|-------------|----------|
| View1 | View2 | View3 | without SCE | with SCE |
| ✓ | ✓ | ✓ | 0.871 | **0.927** |
| ✓ | | | 0.860 | 0.909 |
| | ✓ | | 0.801 | 0.849 |
| | | ✓ | 0.648 | 0.659 |
| ✓ | ✓ | | 0.857 | 0.902 |
| ✓ | | ✓ | **0.873** | 0.921 |
| | ✓ | ✓ | 0.856 | 0.866 |

Table 7: Three views are used for training in the first row, and then selected two of them each time for training, and compared the fusion results.

| num-view | Training Views (without SCE) | | | ACC |
|----------|-------|-------|-------|-------|
| | View1 | View2 | View3 | |
| 3 | ✓ | ✓ | ✓ | 0.873 |
| | ✓ | ✓ | × | 0.861 |
| 2 | ✓ | × | ✓ | 0.856 |
| | × | ✓ | ✓ | 0.707 |

## B.2    Detailed Discussions of View Roles

In this subsection, we conducted further exploration on the UCI-digit dataset to clarify the roles played by different views in cooperation and the effectiveness of cooperation enhancing module. In

Table 8: Three additional datasets summary.

| Dataset | Views | Samples | Clusters |
|---------|-------|---------|----------|
| WikipediaArticles | 2 | 693 | 10 |
| SCENE15 | 2 | 1474 | 7 |
| NoisyMNIST | 2 | 70000 | 10 |

Table 9: Multi-view clustering performance on three additional 2-view benchmark datasets. The optimal results are marked in bold, and the suboptimal values are underlined. O/M denotes out-of-memory error encountered during the training process.

| Methods | WikipediaArticles | | | SCENE15 | | | NoisyMNIST | | |
|---------|------|------|------|------|------|------|------|------|------|
| Metrics | ACC | NMI | ARI | ACC | NMI | ARI | ACC | NMI | ARI |
| DEMVC(2020 IS) | 0.322 | 0.253 | 0.136 | 0.301 | 0.282 | 0.126 | 0.629 | 0.583 | 0.488 |
| CoMVC(2021 CVPR) | 0.261 | 0.176 | 0.158 | 0.239 | 0.287 | 0.118 | 0.278 | 0.265 | 0.095 |
| SiMVC(2021 CVPR) | 0.205 | 0.074 | 0.121 | 0.227 | 0.266 | 0.114 | 0.236 | 0.242 | 0.049 |
| SDSNE(2022 AAAI) | 0.586 | 0.525 | 0.363 | 0.412 | 0.440 | 0.238 | O/M | O/M | O/M |
| MFLVC(2022 CVPR) | 0.429 | 0.346 | 0.288 | 0.389 | 0.409 | 0.217 | 0.990 | 0.975 | 0.979 |
| SDMVC(2022 TKDE) | 0.274 | 0.183 | 0.164 | 0.316 | 0.312 | 0.141 | O/M | O/M | O/M |
| DSMVC(2022 CVPR) | 0.629 | 0.569 | 0.481 | 0.286 | 0.202 | 0.102 | 0.447 | 0.369 | 0.273 |
| APADC(2023 TIP) | 0.491 | 0.391 | 0.338 | **0.445** | 0.449 | 0.268 | 0.694 | 0.801 | 0.682 |
| DMJC(2020 TKDE) | 0.623 | 0.594 | 0.478 | 0.294 | 0.328 | 0.183 | 0.776 | 0.743 | 0.647 |
| DMJC+**SCE** | **0.648** | **0.607** | **0.508** | 0.344 | 0.392 | 0.223 | 0.824 | 0.800 | 0.703 |
| ProIMP(2023 IJCAI) | 0.570 | 0.505 | 0.416 | 0.442 | 0.460 | 0.279 | 0.992 | 0.976 | 0.983 |
| ProIMP+**SCE** | 0.576 | 0.508 | 0.420 | 0.444 | **0.462** | **0.280** | **0.993** | **0.978** | **0.985** |

Table 6, we illustrate the ACC of all views, single view, and paired views after training through three views of UCI-digit. Before we used the view cooperation enhancing module, the fusion results of View1 and View3 were superior to those of all views, although the ACC of View3 as a single view was worst amongst all. Therefore, we consider that views have shortcomings in cooperation. View1 dominates the fusion process, resulting in View2 and View3 not playing a better role. To this end, we used the view cooperation enhancing module to enhance the contributions of View2 and View3.

Furthermore, we compared the results in Table 7 when only two views participated in training. Although the fusion of View1 and View3 showed superior performance when three views participated in training, when training with only View1 and View3, the performance of fusion is significantly lower than that of training with three views. Besides, the fusion performance of paired views is not as good as that of three views, proving that each view in the dataset can play a positive role in cooperation, and this effect can be further expanded by enhancing cooperation.

### B.3 Experiments on Additional Datasets

In this subsection, we conducted experiments on three additional datasets with two views: WikipediaArticles[54], SCENE15[55], and NoisyMNIST[8]. The detailed information about the datasets is shown in Table 8. Considering the scale of the NoisyMNIST dataset, all experiments were conducted on PyTorch platform using the NVIDIA 3090 GPU. The experimental results on these three datasets are presented in Table 9. The experimental results confirm our proposed theory and demonstrate the validity of our View Contribution Evaluation Module as well as the effectiveness of our View Cooperation Enhancing Module.

## C   Limitations and Future Work

In this section, we discussed the limitations of the SCE model and potential future work:

• On some extreme datasets, certain view contributions obtained by the View Contribution Evaluation Module may be negative, which contradicts the non-negativity definition of Shapley values. A

---

[8] http://yann.lecun.com/exdb/mnist/

negative Shapley value indicates that the corresponding view has performed poorly in the fusion process, raising the question of whether these views should be discarded, which could be a potential area of future research.

• The View Cooperation Enhancing Module narrows the contributions between views through gradient modulation, but it cannot extend the contributions between views. Whether there exists a more flexible method to control the contribution relationship between views is an intriguing question.

