# OpenReview forum: "Evaluate then Cooperate: Shapley-based View Cooperation Enhancement for Multi-view Clustering"
_NeurIPS.cc/2024/Conference — NeurIPS 2024 poster_

### Official Review · Reviewer_CDST · 2024-07-08

**Soundness:** 4
**Presentation:** 3
**Contribution:** 3
**Rating:** 6
**Confidence:** 4

**Summary:**

This paper studies multi-view clustering and seeks to investigate the view cooperation issue. The authors consider DMVC as an unsupervised cooperative game and regard each view as a participant. Compared with the existing methods, this consideration is new and interesting. Based on the novel idea, the authors proposed SCE-MVC, a novel shapley-based cooperation enhancing multi-view clustering method. The paper is well-organized. The experiments are convincing.

**Strengths:**

1. The paper proposes a new point also an interesting point for multi-view clustering tasks, i.e., considering the multi-view collaboration as a cooperative game.

2. The experiments are sufficient and convincing. The authors validate the method from many aspects. The proposed SCE-MVC obtains much better performance on six diverse datasets.

**Weaknesses:**

1. Figure 2 is confusing. The specific structure of View Cooperation Enhancing Module is not clearly presented.

2. There are many formulas and symbols. It is suggested to add a notation table.

3. Although the authors try to explain model (1), it is still difficult to understand Shapley Value from the model. In addition, many variables are not clearly explained. The authors should present more information about the model and explain all variables used in this model, such as S_i, {i}, s\{i}， etc.

**Questions:**

The article involves rich theoretical and mathematical knowledge. I have a question to the designation of the model: Which design is the key to improving model performance?

---

> ### Author Rebuttal · Authors · 2024-08-05
>
> **1. Illustration in Figure 2:**
>
> Thanks. Improvements have made to the View Cooperation Enhancing Module in Figure 2 in the Figure 1 of global reponse, presenting the details of gradient modulation within this module.
>
> **2. Notification table:**
>
> Thanks for your suggestions. A notification table will be given in the final version.
>
> **3. Key to improving model performance:**
>
> Thanks. Previous DMVC methods did not adequately consider the aspects of **view cooperation**, especially the joint methods, which may lead to the collapse of fusion results onto one or a few views. Building upon the fundamental assumption of multi-view learning that each view contains complementary information beneficial for downstream tasks, fully leveraging the role of each perspective and enhancing cooperation leads to better model performance.

---

> > ### Comment · Reviewer_CDST · 2024-08-11
> >
> > Thanks for the response. All my concerns have been addressed. After reading all the reviewers' comments and responses, I decided to raise the score.

---

> > > ### Author Response · Authors · 2024-08-12
> > > **Follow Up for reviewer CDST**
> > >
> > > Thank you for your valuable insights on our work. We will eloquently present our idea of view cooperation in the final version.

---

### Official Review · Reviewer_v5TM · 2024-07-08

**Soundness:** 3
**Presentation:** 2
**Contribution:** 3
**Rating:** 6
**Confidence:** 4

**Summary:**

The author introduces a Shapley-based cooperation enhancement framework aimed at fostering collaboration among different views. The SCE-MVC method incorporates cooperative game theory, considering each view as a participant in the model and assessing their contributions using the Shapley Value.

**Strengths:**

Viewing each view as an individual player within game theory represents a fresh perspective in multi-view clustering. Also, enhancing clustering performance through balancing view contribution is both well-founded and innovative.

**Weaknesses:**

1. Using the SCE module in an alignment-based framework only provides a marginal improvement to the model. Does this imply that the SCE module is ineffective in the alignment-based framework?

2. The view contributions of alignment-based method is much balanced than view contributions of joint methods. Does this imply that the alignment-based method is much better than the joint method? It's not reasonable since the clustering performance of alignment-based methods may not necessarily be better than that of joint methods.

3. Is the complexity of computing Shapley values truly O(n!)? When dealing with a larger number of views, can this evaluation framework still be utilized for computation?

4. Are the loss functions L in Eqs (15) and (16) on page 6 the same? If so, there is a problem of inconsistent dependent variables. In addition, $D_ij$ in Eq. (9) is a scalar and should not be bolded.

**Questions:**

The alignment-based method proposed in Theorem 1 will make the contribution values of several views the same. Combined with the experimental results in Table 2, does this mean that the end point of the view contribution optimization proposed in this paper is contrastive learning? If not, please explain in detail the difference between the method in this paper and the contrastive learning method?

**Limitations:**

The authors have adequately addressed the limitations and potential negative societal impact of their work.

---

> ### Author Rebuttal · Authors · 2024-08-05
>
> **1. the use of SCE module on alignment-based methods:**
>
> Thanks. The SCE method consists of two modules: the View Contribution Evaluation Module and the View Cooperation Enhancing Module. For alignment-based methods, the View Contribution Evaluation Module obtains the contributions of views, where these contributions are remarkably close in the alignment-based frameworks, agreeing with the results deduced from our theory.
>
> |    Dataset   |     Method     |  $\phi_1$ |  $\phi_2$ |
> |------------|--------------|-----|-----|
> |      CUB     | InfoNCE+Kmeans | 0.491 | 0.509 |
> |      CUB     | ProIMP         | 0.556 | 0.443 |
> | Caltech101-7 | InfoNCE+Kmeans | 0.484 | 0.516 |
> | Caltech101-7 | ProIMP         | 0.489 | 0.511 |
>
> Building upon the consistent view contributions, the limited enhancements brought by the View Cooperation Enhancing Module can be elucidated. Our experimentation and analysis of alignment-based methods with/without SCE in the paper are not aimed at showcasing the enhancement brought by SCE to the model, but rather at validating the integrity of our theory and the soundness of the View Contribution Evaluation Module.
>
> **2. Is the alignment-based method superior to the joint method?**
>
> Thanks. Given a DMVC framework, the SCE module can evaluate the cooperation among views and make optimizations based on the assessment. If the cooperation among views is insufficient, it indicates potential for improvement upon the original clustering performance, while it does not imply that the original clustering results are poor. For example, on the Caltech101-7 dataset, while DMJC may not match ProImp in view cooperation, it exhibits superior clustering performance. Moreover, the inconsistent view cooperation of DMJC suggests that using the SCE module can yield remarkable enhancements to the model. Our work, from the perspective of view contributions, highlights potential limitations of joint methods, and does not assert that alignment-based methods are superior than joint methods in clustering performance.
>
> | Dataset      | Method | $\phi_1$  | $\phi_2$  | ACC   |
> |--------------|--------|-------|-------|-------|
> | Caltech101-7 | ProIMP | 0.489 | 0.511 | 0.382 |
> | Caltech101-7 | DMJC   | 0.968 | 0.032 | 0.469 |
>
> **3. Algorithmic complexity of SCE module:**
>
> Thanks. SCE module involves computing the fusion distribution for any combination of views. With V views, the complexity of SCE amounts to $O(2^V)$. While the $O(2^V)$ complexity is manageable for a small number of views, it can become cumbersome with a larger number of views. To mitigate the computational burden of calculating Shapley values for numerous views, approximate algorithms can be employed. For instance, the TreeSHAP [1][2] algorithm can compute Shapley values **in polynomial time**, offering a more efficient approach.
>
> **4. Representation of symbols in the formula:**
>
> Thanks. The $L$ in Equation 15 is an abbreviation for $L(\theta^{(v)}_t)$ in Equation 16. The representation of the formulas will be unified in the final version.
>
> **Reference**
>
> [1] Yang J. Fast treeshap: Accelerating shap value computation for trees[J]. arxiv preprint arxiv:2109.09847, 2021.
>
> [2] Muschalik M, Fumagalli F, Hammer B, et al. Beyond TreeSHAP: Efficient Computation of Any-Order Shapley Interactions for Tree Ensembles[C]//Proceedings of the AAAI Conference on Artificial Intelligence. 2024, 38(13): 14388-14396.

---

### Official Review · Reviewer_URj2 · 2024-07-09

**Soundness:** 3
**Presentation:** 3
**Contribution:** 3
**Rating:** 6
**Confidence:** 5

**Summary:**

The study centers on improving task performance via deep multi-view clustering (DMVC) and fostering cooperation among different views. Specifically, the study evaluates view contributions, emphasizing the significance of strengthening cooperation among views.

**Strengths:**

Considering multi-view tasks from a collaborative standpoint represents a novel approach, with the paper's motivation being notably fresh. Moreover, the paper elucidates potential contribution imbalances in the joint method and addresses them through the SCE method, thereby enhancing cooperation among views.

**Weaknesses:**

When dealing with datasets comprising more than two views, such as three views, how can one assess whether the contribution of the views has become more evenly distributed after employing SCE? While the paper visually presents the contributions of the views, could a quantitative method be provided for this evaluation?

**Questions:**

In the unsupervised multi-view scenario, what is the physical meaning of the contribution value of each view proposed in this paper? What is the relationship between the quantitative value of the view's contribution and the clustering performance of a single view ?

**Limitations:**

The authors have adequately addressed the limitations and potential negative societal impact of their work.

---

> ### Author Rebuttal · Authors · 2024-08-05
>
> **1. Quantification of the equilibrium level of view contributions:**
>
> Thanks. Due to the normalized characteristic of view contributions, the cooperation level among views with/without SCE can be compared by calculating the variance of view contributions, denoted as $D(\phi)$. A smaller variance indicates a more consistent contribution among views, reflecting improved cooperation between views.
> The table below illustrates the values of $D(\phi)$ for DMJC with/without SCE. It is evident that using SCE could result in a lower $D(\phi)$ for the views, indicating a better cooperation among them.
>
> |  $D(\phi)$ | Caltech101-7 |  CUB  | UCI-digit | HandWritten | STL10 | Reuters |
> |--------|:------------:|:-----:|:---------:|:-----------:|:-----:|:-------:|
> |   DMJC   |     0.219    | 0.225 |   0.057   |    0.054    | 0.040 |  0.006  |
> | DMJC+SCE |     0.004    | 0.154 |   0.046   |    0.049    | 0.029 |  0.001  |
>
> **2. Physical meaning of view contributions:**
>
> Thanks. In unsupervised multi-view scenario, the physical meaning of view contributions lies in their influence on fusion. The greater the view's contribution, the more significant its influence on the fusion progress.
>
> **3. Relationship between view contribution and view performance:**
>
> Thanks. There is no link between the contribution of a view and its performance. Based on the fundamental assumption of multi-view learning that each view contains complementary information beneficial for downstream tasks, examining the performance of a single view independently is deemed meaningless. Furthermore, it is infeasible to evaluate the quality of a single view under unsupervised conditions. Hence, the aim is to enhance the cooperation among views by ensuring a more balanced view contribution.

---

### Official Review · Reviewer_TWtF · 2024-07-10

**Soundness:** 3
**Presentation:** 3
**Contribution:** 4
**Rating:** 7
**Confidence:** 5

**Summary:**

This research merges game theory with multi-view clustering by introducing the Shapley-based Cooperation Enhancing (SCE) approach. It features a module to systematically evaluate each view's contribution. The approach promotes view cooperation by adjusting the training convergence rate of view parameters based on their contributions. Extensive experiments on various datasets demonstrate the method's effectiveness when applied to different MVC frameworks.

**Strengths:**

1) The paper integrates the Shapley value from game theory into DMVC, allowing for precise assessment of each view's contribution.
2) Theoretical analysis is thorough, with clear and intuitive figures.
3) The manuscript is well-organized and clearly written.

**Weaknesses:**

The article categorizes DMVC into alignment-based and joint methods. What criteria were used for this classification? Furthermore, only one DMJC method is used as a representative for joint methods.

**Questions:**

1) Figure 3(a) indicates that the method does not equalize the contribution value of each view. Why do the contribution values become identical after adding the SCE module to the comparison-based method in Table 2? Please provide a detailed discussion.
2) What criteria were used to classify DMVC?
3) Is DMJC representative of joint methods? Have other joint methods employed similar frameworks?

**Limitations:**

Yes

---

> ### Author Rebuttal · Authors · 2024-08-05
>
> **1. Different characteristics of joint method (Figure 3(a)) and alignment-based method(Table 2):**
>
> Thanks. Figure 3(a) illustrates the change of view contributions of DMJC (a joint method) with/without SCE. In the joint methods, views' representations are optimized in their respective spaces, leading to uneven contributions in the fusion process. Even with the utilization of the SCE module, it can only alleviate the imbalance in view contributions without guaranteeing complete consistency. This is an inherent characteristic of the joint method framework.
>
> |    Dataset   |   Method   |   $\phi_1$   |   $\phi_2$   | ACC |
> |--------------|------------|-------|-------|-------|
> |      CUB     | DMJC       |   0.974  |   0.026  |  0.758  |
> |      CUB     | DMJC+SCE   |   0.892  |   0.108  |  0.797  |
> | Caltech101-7 | DMJC       |   0.968  |   0.032  |  0.469  |
> | Caltech101-7 | DMJC+SCE   |   0.567  |   0.433  |  0.583  |
>
> On the other hand, the two alignment-based methods in Table 2 inherently bring views' representations closer, leading to more consistent contributions between views. With the application of the SCE module, it is possible to achieve essentially consistent view contributions. These experimental results validate the theoretical framework proposed in our study.
>
> |    Dataset   |   Method   |   $\phi_1$   |   $\phi_2$   | ACC |
> |--------------|------------|-------|-------|-------|
> |      CUB     | ProIMP     |   0.556  |   0.443  |  0.825  |
> |      CUB     | ProIMP+SCE |   0.484  |   0.516  |  0.832  |
> | Caltech101-7 | ProIMP     |   0.489  |   0.511  |  0.382  |
> | Caltech101-7 | ProIMP+SCE |   0.499  |   0.501  |  0.382  |
>
> **2. Criteria of classifying DMVC:**
>
> Thanks. Our classification of DMVC methods is inspired by [1], which categorizes multi-view representation learning into (1) joint methods, (2) alignment methods, and (3) shared and specific methods, based on whether the representations of views are brought closer in the same space. In shared and specific methods, shared representations are brought closer to each other in the same space, while specific representations are individually optimized in different spaces, which can be seen as a combination of (1) and (2). Similar classification methods are also found in [2], [3], [4]. Drawing on this classification approach, our work divides DMVC into joint methods, alignment-based methods, and other methods.
>
> **3. Similar frameworks to DMJC:**
>
> Thanks. The methods of DMJC have found numerous applications in MVC research in recent years. Approaches such as SURER [5], SGDMC [6], and DFP-GNN [7] have also employed similar strategies, namely optimizing view representations in respective spaces and using the sharpening of fused distributions as self-supervised signals to guide training.
>
> **Reference**
>
> [1] Jia X, et al. Human collective intelligence inspired multi-view representation learning—Enabling view communication by simulating human communication mechanism[J]. IEEE Transactions on Pattern Analysis and Machine Intelligence, 2022, 45(6): 7412-7429.
>
> [2] Baltrušaitis T, Ahuja C, Morency L P. Multimodal machine learning: A survey and taxonomy[J]. IEEE transactions on pattern analysis and machine intelligence, 2018, 41(2): 423-443.
>
> [3] Li Y, Yang M, Zhang Z. A survey of multi-view representation learning[J]. IEEE transactions on knowledge and data engineering, 2018, 31(10): 1863-1883.
>
> [4] Jia X, et al. Semi-supervised multi-view deep discriminant representation learning[J]. IEEE transactions on pattern analysis and machine intelligence, 2020, 43(7): 2496-2509.
>
> [5] Wang J, Feng S, Lyu G, et al. SURER: Structure-Adaptive Unified Graph Neural Network for Multi-View Clustering[C]//Proceedings of the AAAI Conference on Artificial Intelligence. 2024, 38(14): 15520-15527.
>
> [6] Huang Z, Ren Y, Pu X, et al. Self-supervised graph attention networks for deep weighted multi-view clustering[C]//Proceedings of the AAAI Conference on Artificial Intelligence. 2023, 37(7): 7936-7943.
>
> [7] Xiao S, Du S, Chen Z, et al. Dual fusion-propagation graph neural network for multi-view clustering[J]. IEEE Transactions on Multimedia, 2023, 25: 9203-9215.

---

> > ### Comment · Reviewer_TWtF · 2024-08-11
> >
> > Thank you for your responses, which have addressed my concerns. I am favor of the coopearation idea for MVC and vote for acceptance.

---

> > > ### Author Response · Authors · 2024-08-12
> > > **Follow Up for reviewer TWtF**
> > >
> > > Thank you for your positive comments. Cooperation among views offers a fresh perspective for the DMVC methods.

---

### Official Review · Reviewer_zda3 · 2024-07-12

**Soundness:** 2
**Presentation:** 3
**Contribution:** 3
**Rating:** 6
**Confidence:** 5

**Summary:**

This paper firstly considered DMVC as an unsupervised cooperative game where each view can be regarded as a participant. Then, the authors introduced the shapley value and propose a novel MVC framework termed Shapley-based Cooperation Enhancing Multi-view Clustering (SCE-MVC), which evaluates view cooperation with game theory. In summary, this paper was well written with obvious superiority.

**Strengths:**

-- A MVC framework was designed that utilizeD game theory and Shapley values to evaluate and elevate inter-view cooperation.
-- The experiments were sufficient, and the analysis of the experimental results was adequate.

**Weaknesses:**

-- In this paper, why utilize $\phi_i$ to measure the contribution of views instead of the view weight $w_i$? The article's explanation on this is not clear enough, and there is a lack of experiments to demonstrate the relationship between $\phi_i$ and $w_i$.

**Questions:**

I have the following questions:
-- What will happen if the view contribution is push away from each other?
-- Are there scenarios where narrowing the contribution between views fails to enhance the effectiveness of multi-view clustering?

**Limitations:**

The authors have adequately addressed the limitations and potential negative societal impact of their work.

---

> ### Author Rebuttal · Authors · 2024-08-05
>
> **1. The relationship between $\phi$ and $w$:**
>
> Thanks. Evaluating the view contribution using $\phi$ calculated by shapley value, instead of relying solely on pre-set weights $w$, stems from a systemic perspective on the process of multi-view clustering. In multi-view of representation learning and fusion, conventional static weights ($w$) overlook the intrinsic interactions among views. Contrarily, Shapley values’ distinctive marginal analysis capability allows for precise quantification of each view’s average incremental contribution to the system’s overall performance. Therefore, $\phi$ reveals not only the fundamental worth of each view, but also showcasing the additional value they generate through dynamic interactions.
>
> **2. Pushing away contributions:**
>
> Thanks. Pushing away contributions among different views can be seen as the antithesis of our SCE module. In this context, the fusion of views collapses onto one or a few views, failing to leverage the complementary information of multi-view information.
>
> **3. Counterexamples:**
>
> Thanks. There are situations where the contributions of views are brought closer together without enhancing the clustering performance. The fundamental assumption of DMVC is: Each view contains complementary information beneficial for downstream tasks, better view cooperation leads to better model performance. However, if a dataset obeys this assumption that a certain view contains a significant amount of noise and erroneous information, detrimental to the clustering task, increasing the contribution of that view will result in an overall decrease in clustering performance.

---

> > ### Comment · Reviewer_zda3 · 2024-08-12
> > **Thanks to the reply.**
> >
> > Thanks to your kindly responses. Most of my concerns have been resolved.

---

> > ### Comment · Reviewer_zda3 · 2024-08-13
> > **Thanks for your detailed responses.**
> >
> > I have carefully reviewed the authors' responses. Since all my concerns have been tackled, I understood the motivation, contribution, and experimental analysis more clearly. Combined with the comments from other reviewers, I decided to raise the score of this paper.

---

### Author Rebuttal · Authors · 2024-08-07

We thank the SAC, AC, and PCs for their efforts and constructive comments, which are helpful in further improving the quality of our manuscript. We respond to your questions carefully one by one carefully, and we hope our responses can address your concerns.

Note that there are five tables and one figure in the attached PDF, corresponding to RQ1 for Reviewer TWtF, RQ1 for Reviewer URj2, RQ1 and RQ2 for Reviewer v5TM, and RQ1 for Reviewer CDST.

---

> ### Comment · Reviewer_v5TM · 2024-08-12
>
> Thanks for your responses, my questions have been resolved. Additionally, I checked out the feedback of other reviewers. I think the motivation of this paper commendable. I decide to elevate the rating of the paper to weak accept.

---

> > ### Author Response · Authors · 2024-08-12
> > **Follow Up for reviewer v5TM**
> >
> > Thank you for acknowledging the motivation of our work. The dynamics and cooperation between views is worthy of study.

---

### Author Response · Authors · 2024-08-11

Dear Reviewers, Area Chairs, Senior Area Chairs and Program Chairs,

We sincerely thank the efforts and constructive comments you have made for this paper. The reviewers put forward many insightful questions and valuable suggestions towards improving our paper. In the rebuttal phase, we provided detailed responses to all reviewers' comments point by point, hoping to address the issues raised by reviewers. The discussion period is coming to an end, and we are actively awaiting for further discussion from Reviewers. If you have any other questions, we are happy to discuss them with you at any time. If our response addresses your concerns, please consider increasing your rating. We are looking forward to your reply.

Best,
Authors of paper-781.

---

### Decision · Program_Chairs · 2024-09-25

**Decision:**

Accept (poster)

**Comment:**

This paper treats DMVC as an unsupervised cooperative game, with each view acting as a participant. The authors introduce the Shapley value and propose a novel MVC framework that assesses view cooperation using game theory. Extensive experiments validate the effectiveness of the proposed framework. While some potential issues remain, all reviewers acknowledge the contributions of this paper and have recommended it for acceptance.